# The Influence of Alcohol Consumption on Fighting, Shoplifting and Vandalism in Young Adults

**DOI:** 10.3390/ijerph18073509

**Published:** 2021-03-28

**Authors:** Ieuan Evans, Jon Heron, Joseph Murray, Matthew Hickman, Gemma Hammerton

**Affiliations:** 1Population Health Sciences, Bristol Medical School, University of Bristol, Bristol BS8 2BN, UK; fm19377@bristol.ac.uk (I.E.); jon.heron@bristol.ac.uk (J.H.); Matthew.Hickman@bristol.ac.uk (M.H.); 2Postgraduate Program in Epidemiology, Federal University of Pelotas, Pelotas, Rio Grande do Sul 96020-220, Brazil; j.murray@doveresearch.org; 3Human Development and Violence Research Centre, Federal University of Pelotas, Pelotas, Rio Grande do Sul 96020-220, Brazil; 4MRC Integrated Epidemiology Unit at the University of Bristol, Bristol BS8 2BN, UK

**Keywords:** alcohol, crime, situational decomposition, ALSPAC

## Abstract

Experimental studies support the conventional belief that people behave more aggressively whilst under the influence of alcohol. To examine how these experimental findings manifest in real life situations, this study uses a method for estimating evidence for causality with observational data—‘situational decomposition’ to examine the association between alcohol consumption and crime in young adults from the Avon Longitudinal Study of Parents and Children. Self-report questionnaires were completed at age 24 years to assess typical alcohol consumption and frequency, participation in fighting, shoplifting and vandalism in the previous year, and whether these crimes were committed under the influence of alcohol. Situational decomposition compares the strength of two associations, (1) the *total association* between alcohol consumption and crime (sober or intoxicated) versus (2) the association between alcohol consumption and crime committed while sober. There was an association between typical alcohol consumption and total crime for fighting [OR (95% CI): 1.47 (1.29, 1.67)], shoplifting [OR (95% CI): 1.25 (1.12, 1.40)], and vandalism [OR (95% CI): 1.33 (1.12, 1.57)]. The associations for both fighting and shoplifting had a small causal component (with the association for sober crime slightly smaller than the association for total crime). However, the association for vandalism had a larger causal component.

## 1. Introduction

Drinking alcohol is a practice perceived as normal and integral to many social and cultural occasions throughout the world and across millennia [1,2]. ‘Binge drinking’ among young adults in most countries is often perceived in contemporary youth culture as a ‘social lubricant’ due to its psychoactive effects of reducing anxiety and feeling good [2]. The legal status, ubiquity, and prevalence of the use of alcohol can often mean that the harm to health and damage to society is overlooked and discounted [3], even though the World Health Organization estimates that in 2018 the use of alcohol accounted for 7.1% and 2.2% of the global burden of disease for males and females, respectively [4]. Using data collected through the Alcohol Toolkit Survey in England, 20% of respondents reported experiencing alcohol-related harm from others in the last year, and 5% reported experiencing an aggressive harm (for example, being physically threatened, physically hurt or forced/pressured into something sexual by someone under the influence of alcohol) [5]. An international survey of 63,725 of young people found that approximately 44% of respondents had suffered harm of an aggressive nature from others’ drinking at least once in the last year, with the highest prevalence in those aged 18 to 24 years [6].

It has been proposed that alcohol facilitates crime by causing psychological changes. Affective states may be altered, such as increased arousal and emotional lability, alongside cognitive changes such as risk perception and working memory [7,8,9,10]. It is also possible that alcohol consumption leads to criminal behaviour because of the socio-cultural factors surrounding drinking practices such as the ‘blame it on the bottle’ mentality. It has been suggested that these explanations would result in all types of crime being more likely under the influence of alcohol [11,12]. However, alternative considerations suggest that alcohol intoxication will have stronger effects on crimes that are more serious, that involve personal confrontation and are riskier, and that result from a dispute [11,12]. Some of these explanations have been supported by experimental studies, with reviews documenting that subjects intoxicated with alcohol behave more aggressively in the laboratory compared to subjects that receive placebo substances [13,14]. A review of meta-analyses of experimental studies supported the view that alcohol produces certain cognitive changes that interact with external factors (such as frustration) to increase the likelihood of aggression [13]. These findings support the conventional belief that people behave more aggressively whilst under the influence of alcohol, but such experimental findings are difficult to corroborate with observational studies ‘in the field’ [15].

Many observational studies have found strong associations between alcohol consumption and crime, but the extent to which these associations are causal is unclear [15]. Associations may be confounded by factors such as low self-control, propensity for risk taking, socio-demographic factors, deviant peer groups and compromised domestic and family contexts. Even when observational studies have measured potential confounders extensively, there could still be some bias from residual confounding because of other unmeasured confounders, or because of measurement error. Felson and colleagues developed an approach named ‘situational decomposition’ to help overcome this problem, and have conducted several observational studies on the relationship between alcohol consumption and various behaviours in adolescents [12,16,17]. Situational decomposition assumes that the *total association* between alcohol consumption and crime can be broken down into two contributing components, non-causal (referred to as ‘spurious’ in previous literature) and causal. The method attempts to identify the causal effects of alcohol on crime by comparing the relative strength of two associations—the association between alcohol consumption and sober crime and the association between alcohol consumption and any crime (either sober or under the influence of alcohol) [12]. The association between alcohol consumption and committing a crime whilst sober represents the confounded part of the association, as it cannot be due to the situational effects of intoxication. It therefore quantifies the influence of potential confounders for alcohol consumption and crime (such as low self-control, socio-demographic factors and compromised domestic and family contexts). For example, if those that drink heavily are more likely to commit a crime when they are sober, this cannot be due to the causal effects of intoxication, and therefore must be due to common causes of heavy alcohol use and crime. It is assumed that the total association between alcohol consumption and crime reflects both confounding and the causal effect of alcohol on crime [16]. Situational decomposition infers the ‘causal effect’ of alcohol on crime as the difference between the total association and the confounded association. A total association approximately equal to the confounded association would indicate no causal effect. The smaller the size of the confounded association (sober crime) relative to the total association, the larger the relative size of the causal effect (due to the situational effects of intoxication) [16]. Drawing causal conclusions from observational studies is extremely challenging, and no single study or method can provide a definite answer to a causal question. However, the method of situational decomposition is less prone to biases from measurement error and residual confounding compared to traditional analyses using observational data, given the focus on comparing the relative size of coefficients [12]. We use the terms ‘causal’ and ‘non-causal’ to describe the associations in the current paper for consistency with previous literature using situational decomposition, but also to provide a clear distinction to the part of the association with crime which reflects only confounding (the association with sober crime), and the remainder of the association. However, we acknowledge that there are threats to causality using any analytical approach with observational data. 

Felson and colleagues have used situational decomposition to examine the influence of alcohol consumption on committing various types of crime in a sample of Finnish [12] and American adolescents [16]. Using the sample of Finnish adolescents, Felson and colleagues concluded that the association between alcohol frequency and ‘covert’ or ‘petty’ crime (e.g., shoplifting or stealing from home) only had a very small causal component. However, for the association between alcohol frequency and ‘anti-social’ or ‘conspicuous’ crime (e.g., violence, vandalism, car theft, and graffiti), stronger evidence for causality was present [12]. The small causal relationship for ‘petty’ crime may be explained by a lack of opportunity for both alcohol consumption and committing these types of crime, or a lower level of impulsivity associated with these crimes [12]. Using the sample of American adolescents, Felson and colleagues found that the association between alcohol prevalence and fighting was mostly non-causal; however, there was a stronger causal component for the association for fighting with both alcohol frequency and the amount of alcohol consumed in a typical drinking session [16]. This study also found that the amount of alcohol consumed had a strong association with fighting under the influence of alcohol (versus no fighting), also suggesting a causal relationship. It is not possible to interpret the association between alcohol frequency and crime under the influence of alcohol because a frequent drinker is more likely to be under the influence of alcohol than an infrequent drinker on any occasion, not only when committing a crime. However, the association for the amount of alcohol consumed can be interpreted, if alcohol frequency is controlled for in the analyses [16]. 

In the present study, we aim to extend the previous literature by examining the evidence for a causal association between the amount of alcohol typically consumed and three types of crime (fighting, shoplifting and vandalism). We will examine whether the findings generalize to young adults in the UK by using a population-based sample of 24-year-olds (from the Avon Longitudinal Study of Parents and Children). Investigating whether there is consistency in findings across different demographic groups and cultures is important as it helps to strengthen conclusions about causal relationships. However, it is possible that the causal component of associations between alcohol consumption and crime may be larger in young adult (compared to adolescent) samples, given that drinking is legal, therefore reducing the importance of individual differences (such as a delinquent peer group, lack of self-control, and impulsivity) that may lead to general delinquent behaviour [16]. Based on previous findings using adolescent samples, we hypothesise that the association between alcohol consumption and shoplifting will be mostly non-causal, whereas the association with fighting and vandalism will show some evidence for causality. We will also examine evidence for an interaction between sex and alcohol consumption on crime, given findings from previous studies that alcohol has a stronger influence on crime in males compared to females [11,18,19]. 

## 2. Materials and Methods 

### 2.1. Sample

Data were utilised from a large UK birth cohort, the ‘Avon Longitudinal Study of Parents and Children’ (ALSPAC) which was set up to examine genetic and environmental determinants of health and development [20]. ALSPAC recruited pregnant women resident in Avon, UK with expected dates of delivery from 1 April 1991 to 31 December 1992. Of these initial 14,541 pregnancies, there was a total of 14,676 foetuses, resulting in 14,062 live births and 13,988 children who were alive at 1 year of age. Parents and children have been followed up regularly since recruitment via questionnaire and clinic assessments. Study data from 2014 onwards were collected and managed using REDCap electronic data capture tools hosted at University of Bristol [21,22]. Further details on the sample characteristics and methodology have been described previously [20,23,24], and the study website contains details of all the data that is available through a fully searchable data dictionary and variable search tool (http://www.bristol.ac.uk/alspac/researchers/our-data/). Of the 9292 of ALSPAC participants who were invited for a clinical assessment at approximately age 24 years, 3726 attended (mean age of 24 years (22 to 26 years); 1407 males, 2319 females). Table 1 compares the original ALSPAC sample to those that attended the clinical assessment at age 24 years (*n* = 3726) on sociodemographic and parental characteristics. All analyses were restricted to those who reported that they had consumed alcohol in the previous year (*n* = 3408), given that it is not possible for an individual who abstains to have committed crime under the influence of alcohol. See Appendix A for a comparison of sociodemographic characteristics for those that have (*n* = 3408) and have not (*n* = 166) consumed alcohol in the past year.

### 2.2. Ethical Considerations and Data Availability

Ethical approval for the study was obtained from the ALSPAC Ethics and Law Committee (IRB00003312) and the Local Research Ethics Committees. Informed consent for the use of data collected via questionnaires and clinics was obtained from participants following the recommendations of the ALSPAC Ethics and Law Committee at the time.

The data underlying this study are third party data. Data used for this submission will be made available on request to the ALSPAC executive committee (alspac-exec@bristol.ac.uk). The ALSPAC data management plan (available here: http://www.bristol.ac.uk/alspac/researchers/data-access/) describes in detail the policy regarding data sharing, which is through a system of managed open access.

### 2.3. Measures

Data were assessed using computerized self-report questionnaire measures administered during a clinical assessment at approximately at 24 years. 

#### 2.3.1. Alcohol Consumption

Typical amount of alcohol consumed and alcohol frequency were assessed using two items from the self-report Alcohol Use Disorders Identification Test (AUDIT [25]) which is a brief screening tool to identify individuals with alcohol-related problems. The AUDIT scale has been studied extensively and has been validated using diagnostic interview, physical examinations and laboratory testing [26]. To assess typical consumption, respondents were asked to report how many units containing alcohol they have on a typical day when they are drinking, with responses classified into five categories: ‘1 or 2 units,’’ “3 or 4 units,’’ ‘’5 or 6 units’’, ‘’7 to 9 units’’ and “10 or more units”. To assess frequency, respondents were asked to report how often in the past year they had a drink containing alcohol, with responses classified into five categories: ‘’never,’’ “monthly or less,’’ ‘’two to four times a month’’, ‘’two to three times a week’’ and “four or more times a week”. Typical consumption was the exposure of interest, and alcohol frequency was treated as a confounder in the main analyses. Associations between alcohol frequency and crime were examined in a sensitivity analysis. Both measures of alcohol consumption were treated as numeric in the analyses.

#### 2.3.2. Criminal Behaviour

Criminal behaviour was assessed using a self-report questionnaire referring to crimes committed in the previous year, originally developed in the Edinburgh Study of Youth Transitions and Crime [27]. External validity for this self-report questionnaire has been examined previously in adolescents using crosschecks with official records and teachers’ questionnaires [28]. The current study was concerned with three categories of crime committed in the previous year: fighting “…have you hit, kicked, or punched someone else on purpose with the intention of really hurting them?”, shoplifting “…have you stolen something from a shop or store?”, and vandalism “…have you deliberately damaged or destroyed property that did not belong to you?”. To assess the involvement of alcohol in each type of crime, respondents were asked whether they were under the influence of alcohol when they committed the crime, with responses classified into two categories: “never”, “at least once”. For the purposes of data analysis, three nominal variables were created for each type of crime with response categories: “did not commit crime”, “committed crime at least once, but never under the influence of alcohol”, “committed crime under the influence of alcohol at least once”.

### 2.4. Statistical Analysis

The prevalence of each type of alcohol use (frequency and typical consumption), each type of crime (fighting, shoplifting and vandalism), and whether each crime was committed under the influence of alcohol are reported for the full sample and stratified by sex. The chi-square test of independence was performed to examine difference in prevalence for males and females.

The total association between typical alcohol consumption and each type of crime (fighting, shoplifting and vandalism) was examined using a binary logistic regression. The association between alcohol consumption with sober crime, and crime under the influence of alcohol was tested using multinomial logistic regression. The reference category for both models was ‘did not commit crime’. A multinomial logistic regression is a simple extension of a binary logistic regression that allows for more than two categories of the outcome. For an outcome with three categories, the multinomial regression model estimates two logit equations with one category (did not commit crime) selected as the reference group to compare with the others. Results are presented as multinomial log odds ratios, and multinomial odds ratios. A ratio variable, representing percentage of the total association explained by confounding (referred to as percentage spuriousness for consistency with previous studies), was calculated by dividing the multinomial log odds ratio for sober crime by the log odds ratio for total crime, and multiplying by 100. The multinomial logistic regressions were repeated with sober crime as the reference category to enable direct comparison of crime under the influence of alcohol to sober crime.

All associations for typical alcohol consumption were adjusted for alcohol frequency, given that a person who drinks more often, is more likely to be intoxicated whilst committing a crime purely due to a higher number of opportunities for this to happen [16]. No other confounders were adjusted for in the analyses given that potential confounders are captured in the estimate for the association between alcohol consumption and sober crime, and the focus of the study was on comparing the relative magnitude of this association to the total association between alcohol and crime. Interactions between sex and alcohol consumption on each type of crime were examined. 

Finally, a sensitivity analysis was performed examining the association between alcohol frequency with total crime and sober crime. The association between alcohol frequency and crime under the influence of alcohol was not examined given that these results are difficult to interpret. This is because an individual who consumes alcohol frequently is more likely to be under the influence of alcohol at any given time, not only when committing a crime. Analyses were performed using Stata version 15.0.

## 3. Results

### 3.1. Prevalence of Crime and Alcohol Consumption

Table 2 displays the prevalence of each type of crime (fighting, shoplifting and vandalism), whether it was committed under the influence of alcohol, and the alcohol phenotypes (frequency and typical consumption). In the full sample, the prevalence of fighting, shoplifting and vandalism in the last year was 4%, 6% and 2%, respectively. The prevalence for committing the crime sober or under the influence of alcohol was the same for fighting (both 2%) and vandalism (both 1%) but shoplifting sober was more common than shoplifting under the influence of alcohol (5% versus 1%).

The prevalence of fighting (7% vs. 3%; *p* < 0.001) and vandalism (4% vs. 1%; *p* < 0.001) was higher in men compared to women. However, there was no evidence to suggest the prevalence of shoplifting differed between men and women (6% vs. 6%, *p* = 0.188). Men and women differed on their frequency of alcohol consumption (*p* < 0.001); for example, 8% of men drank four or more times a week compared to 4% of women. Men also drank more than women in a typical session (*p* = 0.012). For example, 13% of men drank 10 or more units in a typical drinking session compared to 11% of women.

Of those who had been in a fight only whilst sober (*n* = 61), 26% had been in a fight more than once in the previous year; whereas, of those who had been in a fight under the influence of alcohol (*n* = 78), 35% had been in a fight more than once. Of those who shoplifted only whilst sober (*n* = 179), 49% had shoplifted more than once in the previous year, compared to 50% of those who had shoplifted under the influence of alcohol (*n* = 24). Of those who vandalized only whilst sober (*n* = 38), 24% had vandalized more than once in the previous year, compared to 28% of those who had vandalized under the influence of alcohol (*n* = 40).

### 3.2. Associations between Typical Alcohol Consumption and Each Type of Crime

Table 3, Model 1 displays results from the binary logistic regression showing the association between typical alcohol consumption (treated as a numeric variable) and total crime (versus no crime); whereas Table 3, Model 2 displays results from the multinomial logistic regression showing the association between typical alcohol consumption and both crime while sober (versus no crime), and crime under the influence of alcohol (versus no crime). All analyses are adjusted for alcohol frequency allowing the association with crime under the influence of alcohol to be interpreted.

There was an association between typical alcohol consumption and total crime (whether sober or under the influence of alcohol) for fighting [OR (95% CI): 1.47 (1.29, 1.67)], shoplifting [OR (95% CI): 1.25 (1.12, 1.40)], and vandalism [OR (95% CI): 1.33 (1.12, 1.57)]. For example, the odds of fighting is multiplied by nearly 1.5 when the amount of alcohol typically consumed increases by one category. There was a small causal component to the associations for both fighting and shoplifting (with the association for sober crime slightly smaller than the association for total crime). However, the association for vandalism had a larger causal component. For example, the log odds ratio for typical alcohol consumption and sober vandalism (representing the non-causal association) was 15 percent as high as the log odds ratio for total vandalism (which combines both the non-causal and causal association). The differences in strength for the associations with total crime, and sober crime are displayed visually in Figure 1.

For all three types of crime, the association with committing the crime under the influence of alcohol was stronger than the association with committing the crime sober, although the confidence intervals for the two associations overlapped for fighting. The multinomial logistic regressions were repeated with sober crime as the reference category to enable direct comparison of crime under the influence of alcohol to sober crime. Typical alcohol consumption was associated with increased odds of committing crime under the influence of alcohol (versus sober crime) for fighting [OR (95% CI): 1.32 (1.03, 1.70)], shoplifting [OR (95% CI): 1.81 (1.27, 2.59)], and vandalism [OR (95% CI): 1.66 (1.19, 2.33)].

There was no evidence for an interaction between sex and typical alcohol consumption in the binary logistic regression for fighting [OR (95% CI): 1.09 (0.84, 1.41)], shoplifting [OR (95% CI): 1.03 (0.83, 1.28)], and vandalism [OR (95% CI): 0.83 (0.58, 1.19)] nor in the multinomial logistic regressions for any type of crime (full results available on request). 

### 3.3. Associations between Alcohol Frequency and Each Type of Crime

As a sensitivity analysis, we examined the association between alcohol frequency (treated as a numeric variable) and crime. Table 4, Model 1 displays results from the binary logistic regression showing the association between alcohol frequency and total crime (versus no crime); whereas Table 2, Model 2 displays results from the multinomial logistic regression showing the association between alcohol frequency and crime whilst sober (versus no crime).

There was little evidence for an association between alcohol frequency and total crime (whether sober or under the influence of alcohol) for fighting [OR (95% CI): 1.21 (0.99, 1.48)] and no evidence for vandalism [OR (95% CI): 1.14 (0.88, 1.48)]. However, there was a weak association between alcohol frequency and shoplifting [OR (95% CI): 1.29 (1.09, 1.52)], which appeared to be partly causal. For example, the log odds ratio for alcohol frequency and sober shoplifting (representing the spurious association) was 41 percent as high as the log odds ratio for total shoplifting (which combines both the spurious and causal association). The percentage of spuriousness could not be calculated for fighting and vandalism due to a negative association between alcohol frequency and committing these crimes whilst sober (for example, drinking more frequently was associated with lower odds of vandalizing whilst sober).

## 4. Discussion

This current study applied the method of situational decomposition to evaluate the extent to which the association between alcohol consumption and crime is causal in a general population sample of young adults. There was an association between typical alcohol consumption and total crime (whether sober or under the influence of alcohol) for fighting, shoplifting and vandalism. The associations for both fighting and shoplifting appeared to be largely explained by confounding (with 63 percent of the association being non-causal for fighting, and 73 percent for shoplifting). However, the association for vandalism had a larger causal component (with just 15 percent of the association being non-causal). For all three types of crime, the association between typical alcohol consumption and committing crime under the influence of alcohol was stronger than the association with committing crime sober, after adjusting for frequency of alcohol consumption, suggesting some causal component. These findings suggest that the amount of alcohol consumed by young adults has a modest impact on whether they engage in fighting and shoplifting, and a larger impact on whether they vandalize. Inferring causation with observational data is always tentative, and this study is no exception; however, it does provide some insight into the situational effects of alcohol on crime. When examining alcohol frequency as a sensitivity analysis, there was little evidence of an association with total crime (except for a weak association for shoplifting which appeared to be partly causal), and no evidence that frequency of alcohol use increased the odds of committing a crime whilst sober.

This study has several strengths including the use of a large population-based sample (ALSPAC) with well-validated measures of both alcohol consumption and criminal behaviour, meaning that we were able to examine the association between alcohol consumption with a diverse range of crimes. We also used a methodology—situational decomposition—which is less prone to biases from measurement error and residual confounding compared to traditional analyses using observational data, given the focus on comparing the relative size of coefficients. We extend the previous studies using this method to examine the relationship between alcohol and crime by using a UK-based sample of young adults, and by examining both multiple types of alcohol consumption (allowing the association with crime under the influence of alcohol to be interpreted) and multiple types of crime (fighting, shoplifting and vandalism). 

However, the results need to be considered in the context of the limitations. First, given that the primary interest of this study was comparing the strength of the association with total crime to the strength of the association with sober crime, it is important to consider any forms of bias that may have impacted on these associations differently. It is possible that the assessment of total crime may be more subject to recall bias than the assessment of sober crime, given that a respondent may forget an incident of crime under the influence of alcohol due to alcohol-induced memory loss [29]. This bias could therefore weaken the total association, but not influence the sober association resulting in the percentage of spuriousness being overestimated. Additionally, as with most cohort studies, there was selective attrition over time with only a small proportion of the original sample attending the clinical assessment at age 24 years. The analysis sample was more affluent than the original sample due to attrition, and there is a strong socioeconomic gradient in committing crime. Although this is likely to bias the absolute strength of associations examined, it should not impact on the relative strength (total association compared to sober association) given that missing data mechanisms are unlikely to be different for total crime versus sober crime. However, generalizability of the findings will be affected. Additionally, only a small proportion of the sample had committed each crime in the previous year meaning that a lack of power may still have impacted on our ability to detect associations. It is also important to note that the absolute size of the association between alcohol consumption and total crime should not be interpreted causally due to potential bias from both confounding and attrition which were not addressed as the absolute size of this association was not the focus of the current study.

Second, a source of heterogeneity that cannot be ruled out from this method comes from unobserved contextual effects that might be related to both drinking and criminal behaviour. For example, alcohol is typically consumed socially, and peer group influences are likely to impact on committing a crime. Certain noisy and crowded environments may also encourage heavy drinking, and engagement in criminal activity such as fighting. Additionally, alcohol consumption often happens at night-time when the streets tend to be dark and quiet. The increased anonymity that night-time provides could encourage vandalism. These situational factors are less likely to be present for the association between alcohol consumption and shoplifting because drinking is usually a night-time activity when most shops are shut. These examples of unobserved contextual effects could have biased our results; however, it is important to consider the possibility that the effects of alcohol are sparking desire in people to seek situations that favor criminal behaviour. In which case these situational factors may in fact be mediators, rather than confounders. Third, the method used in this study specifically examines the effect of an individual’s own drinking on crime, rather than spill-over effects of drinking on other people’s crime. Future research on this topic would benefit from greater specificity, particularly when examining fighting. Additional information such as, who was involved (friends, stranger, partner, or family), who was the victim/aggressor, where it happened, and why it happened would provide greater insight into the causal mechanism of interest.

In the current study, the association between typical alcohol consumption and shoplifting was found to be mostly non-causal, although there was evidence for a small causal component. This finding supports a previous study in Finnish adolescents that found the association between alcohol frequency and shoplifting was almost entirely non-causal (approximately 90%). The small causal relationship between alcohol and shoplifting may be explained by a lack of opportunity for both alcohol consumption and shoplifting, or a lower level of impulsivity associated with shoplifting [12]. Somewhat surprisingly, the association between alcohol frequency and shoplifting in the current study showed a slightly larger causal component than the association for typical consumption. 

Our findings were also similar to previous studies for fighting with just over half of the association being found to be non-causal in both the current study and the previous study in Finnish adolescents [12]. Additionally, another previous study by Felson and colleagues, using American adolescents, found that the association between typical alcohol consumption and fighting had a strong causal component [16]. In the current study, the strongest causal component was found for the association between typical alcohol consumption and vandalism, with no association between alcohol consumption and sober vandalism. This finding is somewhat similar to the study in Finnish adolescents by Felson and colleagues, where just under half of the association was found to be causal [12]. The stronger causal component found in our study could be explained by the alcohol phenotype used (typical alcohol consumption has been shown to have a stronger causal effect than alcohol frequency; [16]), or the age of the sample (with stronger causal effects hypothesised in young adults when drinking is legal, therefore reducing the importance of confounders such as peer deviance and lack of self-control; [16]). In the current study, we found little evidence for an association between alcohol frequency and total crime for fighting and vandalism, and no evidence that frequency of alcohol use increased the odds of committing a crime whilst sober. This finding could be due to frequent drinking being fairly normative in young adulthood, and because those that drink very frequently tend to consume a smaller quantity of alcohol in a typical session. Therefore, the association for frequency is less likely to be due to potential confounders such as peer deviance or a lack of self-control. More research is needed in this age group using a variety of alcohol phenotypes and types of crime. 

We also found that although the prevalence of both crime and alcohol consumption differed by sex, the associations did not. This is in contrast to some previous studies that have reported a stronger influence of alcohol on crime in males compared to females [11,18,19]. 

This study adds to a growing body of literature supporting a short-term effect of alcohol consumption on crime using a variety of methods. Reviews of experimental studies show that subjects intoxicated with alcohol behave more aggressively in the laboratory compared to subjects that receive placebo substances [13,14]. Additionally, observational studies have shown within-person effects of alcohol consumption on antisocial or criminal behaviour in young adulthood [30,31,32,33], indicating that when a young person reports consuming more alcohol than normal, they also report engaging in higher than usual levels of antisocial behaviour. Norström and Pape, 2010 used a similar method (a type of fixed effects modelling) in a Norwegian sample and found that changes in heavy drinking were associated with changes in fighting across adolescence and young adulthood, particularly in those with suppressed anger [34]. These study designs can eliminate the effects of time-stable confounders, therefore strengthening causal inference. Finally, research using a nationally representative sample of inmates in the United States found that alcohol intoxication was more strongly related to crimes that involved personal confrontation, such as homicide, physical and sexual assault, and robbery compared to crimes such as theft and drug offenses [11].

Situational decomposition has been used previously to examine the effects of alcohol on a broader range of outcomes including victimisation [35,36], and sexual intercourse and contraception use [17]. Future research could also use this method to examine exposures other than alcohol use, such as cannabis or other illegal drug use. More research is also needed to examine what factors are associated with committing crimes under the influence of alcohol in young adults. Previous studies of adolescents have found that alcohol intoxication has stronger effects on those who are older, white and who already have violent tendencies [16], and those who were more impulsive and had more deviant peers, after accounting for levels of alcohol use [15].

## 5. Conclusions

This study used a method for estimating causation with observational data—‘situational decomposition’ to examine the association between typical alcohol consumption and crime in young adults in a UK-based general population sample. The associations for both fighting and shoplifting had a small causal component (with the association for sober crime slightly weaker than the association for total crime). However, the association for vandalism had a larger causal component. There are threats to causality using any analytical approach with observational data, and therefore these results need to be interpreted cautiously. However, the findings provide preliminary evidence that crime prevention strategies in young adults should broaden their focus to address heavy alcohol consumption to reduce crime, particularly for vandalism. Future research should consider the context in which a reduction in alcohol consumption could lead to a reduction in crime.

## Figures and Tables

**Figure 1 ijerph-18-03509-f001:**
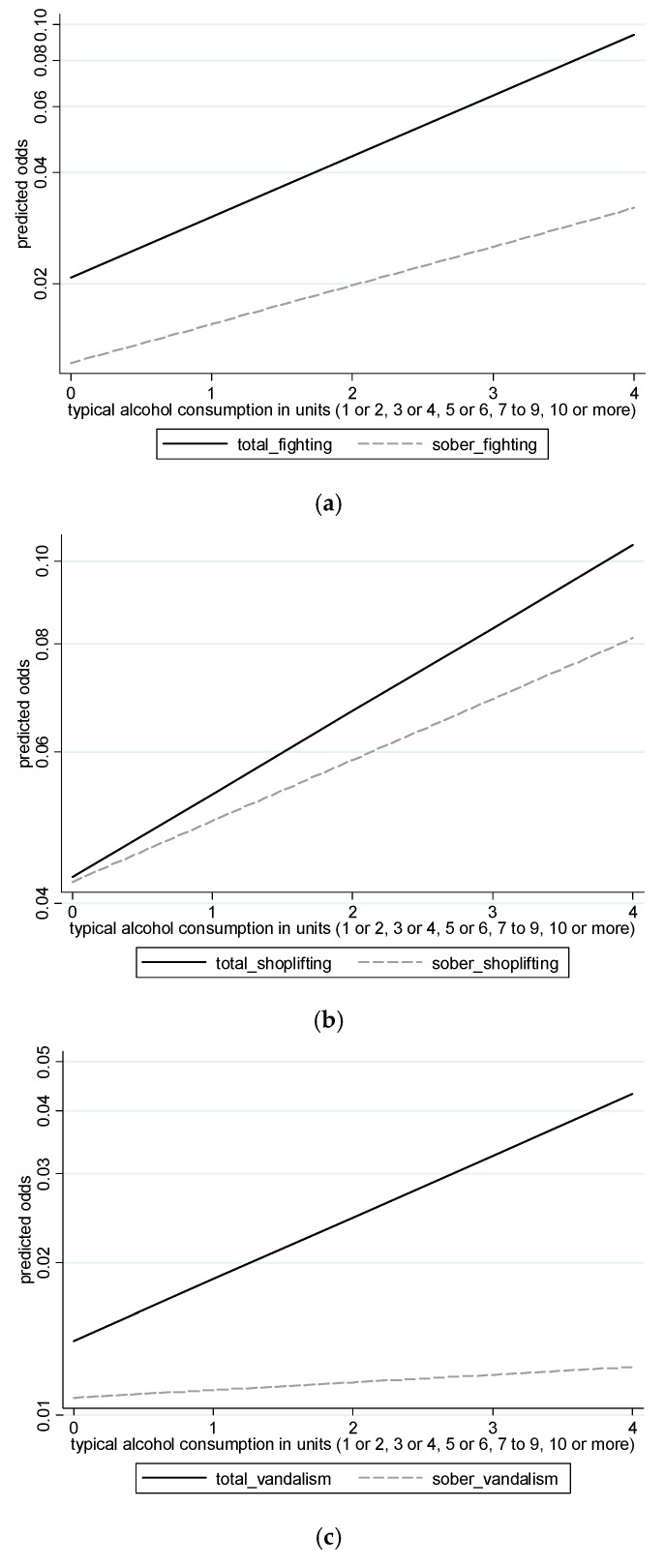
(**a**) Predicted odds of total fighting (solid black line) and sober fighting (dashed grey line) according to units of alcohol consumed in a typical drinking session; (**b**) Predicted odds of total shoplifting (solid black line) and sober shoplifting (dashed grey line) according to units of alcohol consumed in a typical drinking session; (**c**) Predicted odds of total vandalism (solid black line) and sober vandalism (dashed grey line) according to units of alcohol consumed in a typical drinking session. In each figure, the solid black line represents the total association (both non-causal and causal), and the dashed grey line represents the non-causal association; a black line steeper than the grey line indicates a causal effect of alcohol on crime.

**Table 1 ijerph-18-03509-t001:** Comparison of sociodemographic and parental characteristics for the original ALSPAC sample (*n* ≤ 13,960) and those that attended the clinical assessment at age 24 years (*n* ≤ 3726).

	Original ALSPAC Sample		Attended Clinic at 24 Years	
	%	*n*	%	*n*
Sex (male)	52	7209	38	1407
Ethnicity (white)	97	11,990	98	3526
Maternal education (beyond high school)	35	4378	49	1774
Housing tenure (mortgaged)	73	9529	85	3079
Parental crime across childhood (yes)	13	1575	13	464
Parental alcoholism across childhood (yes)	7	894	7	267

**Table 2 ijerph-18-03509-t002:** Prevalence of each type of crime (fighting, shoplifting and vandalism) and whether it was committed under the influence of alcohol and alcohol consumption phenotypes.

	Full Sample		Males		Females	
	%	*n*	%	*n*	%	*n*
**Fighting at least once in past year**						
Yes, under the influence of alcohol at least once	2	78	4	49	1	29
Yes, never under influence of alcohol	2	61	3	33	1	28
Did not commit crime	96	3256	94	1192	97	2064
**Shoplifting at least once in past year**						
Yes, under the influence of alcohol at least once	1	24	1	14	0.5	10
Yes, never under influence of alcohol	5	179	6	71	5	108
Did not commit crime	94	3197	93	1191	94	2006
**Vandalism at least once in past year**						
Yes, under the influence of alcohol at least once	1	40	3	34	0.3	6
Yes, never under influence of alcohol	1	38	1	17	1	21
Did not commit crime	98	3319	96	1218	99	2101
**Frequency of alcohol consumption in past year**						
Monthly or less	23	779	16	207	27	572
Two to four times a month	40	1362	38	481	41	881
Two to three times a week	32	1076	38	482	28	594
Four or more times a week	6	191	8	107	4	84
**Typical consumption of alcohol in the past year**						
1 or 2 units	22	742	21	268	22	474
3 or 4 units	32	1092	29	374	34	718
5 or 6 units	21	710	22	276	20	434
7 to 9 units	14	462	14	185	13	277
10 or more units	12	396	13	171	11	225

Only includes respondents who reported that they had consumed alcohol in the previous year (*n* = 3408).

**Table 3 ijerph-18-03509-t003:** Multivariable associations between typical alcohol consumption (treated as a numeric variable) and total crime (model 1), sober crime (model 2), and crime under the influence of alcohol (model 2).

	Model 1: Binary Logistic		Model 2: Multinomial Logistic				
	Total Crime (Versus No Crime)		Sober Crime (Versus No Crime)		Intoxicated Crime (Versus No Crime)		
	Log OR	OR (95% CI)	Log OR	OR (95% CI)	Log OR	OR (95% CI)	Spuriousness
Fighting	0.38	1.47 (1.29, 1.67)	0.24	1.27 (1.05, 1.53)	0.51	1.67 (1.40, 1.99)	63%
Shoplifting	0.23	1.25 (1.12, 1.40)	0.16	1.18 (1.05, 1.32)	0.76	2.14 (1.52, 3.01)	73%
Vandalism	0.28	1.33 (1.12, 1.57)	0.04	1.04 (0.82, 1.33)	0.55	1.73 (1.35, 2.21)	15%

Only includes respondents who reported that they had consumed alcohol in the previous year (*n* = 3408), actual *n*s vary from 3389 (fighting) to 3394 (shoplifting); model 1 is performed using a binary logistic regression and shows (log) odds ratio and 95% confidence interval for the total association between typical alcohol consumption and crime (versus no crime); model 2 is performed using a multinomial logistic regression and shows multinomial (log) odds ratio and 95% confidence interval for the association between typical alcohol consumption and both sober crime (versus no crime), and crime under the influence of alcohol (versus no crime); analyses adjusted for alcohol frequency; spuriousness was calculated by diving the log odds ratio for sober crime (model 2) by the log odds ratio for total crime (model 1), and multiplying by 100.

**Table 4 ijerph-18-03509-t004:** Univariable associations between alcohol frequency (treated as a numeric variable) and total crime (model 1), and sober crime (model 2).

	Model 1: Binary Logistic		Model 2: Multinomial Logistic		
	Total Crime (Versus No Crime)		Sober Crime (Versus No Crime)		
	Log OR	OR (95% CI)	Log OR	OR (95% CI)	Spuriousness
Fighting	0.19	1.21 (0.99, 1.48)	−0.29	0.74 (0.55, 1.01)	NA
Shoplifting	0.25	1.29 (1.09, 1.52)	0.10	1.11 (0.93, 1.32)	41%
Vandalism	0.13	1.14 (0.88, 1.48)	−0.52	0.60 (0.40, 0.89)	NA

Only includes respondents who reported that they had consumed alcohol in the previous year (*n* = 3408), actual *n*s vary from 3395 (fighting) to 3400 (shoplifting); model 1 is performed using a binary logistic regression and shows (log) odds ratio and 95% confidence interval for the total association between alcohol frequency and crime (versus no crime); model 2 is performed using a multinomial logistic regression and shows multinomial (log) odds ratio and 95% confidence interval for the association between alcohol frequency and sober crime (versus no crime); spuriousness was calculated by diving the log odds ratio for sober crime (model 2) by the log odds ratio for total crime (model 1), and multiplying by 100.

## Data Availability

The data underlying this study are third party data. Data used for this submission will be made available on request to the ALSPAC executive committee (alspac-exec@bristol.ac.uk). The ALSPAC data management plan (available here: http://www.bristol.ac.uk/alspac/researchers/data-access/) describes in detail the policy regarding data sharing, which is through a system of managed open access.

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
