# Peer review of "The Influence of Alcohol Consumption on Fighting, Shoplifting and Vandalism in Young Adults"

_ijerph, 2021, doi:10.3390/ijerph18073509_

Round 1
Reviewer 1 Report
I think this research is well-done and the paper is well-written. I only have some minor criticisms.
Clarify: “Binge drinking’ among young adults in most countries is normalised through marketing tactics.”
There has been plenty of attention paid to the issue of how alcohol is related harm to others.
Sensitivity analysis: you should analyze the data using frequency of drinking to see if the results are the same. Also, what do the results look like when you don’t restrict the analysis to those who reported that they had consumed alcohol in the previous year? I don’t think that is necessary to omit non-drinkers. Previous research using this method did not.
Restate: “There was little evidence to suggest the prevalence of shoplifting differed between men and women.” There was no evidence.
The heading for table 3 is too long. Some of that material goes in the note at the bottom.
The following statement is inaccurate: “The associations for both fighting and shoplifting appeared to be largely non-causal (with the association for sober crime similar in strength to the association for total crime).” You do find evidence of causality for fighting and shoplifting. You also misstate this elsewhere (e.g., “The lack of evidence for causality for ‘petty’ crime…”).
I’m not sure I understand the sensitivity analysis where you “replicated the results using two binary variables for each type of crime, instead of one nominal variable.” You might want to skip it.
Say a little more about the sex differences. You say that a lack of power may have impacted your ability to detect sex interactions. What makes you think lack of power was responsible? Were the odds ratios somewhat different? Was the difference close to significant?
“This finding may be explained by a lack of opportunity for both alcohol consumption and shoplifting, or a lower level of impulsivity associated with shoplifting.” Doesn’t the method control for time-varying confounders?
The following seems inaccurate: “In the current study, the strongest causal component was found for the association between typical alcohol consumption and vandalism, with no association between alcohol consumption and sober vandalism. This finding is in contrast to the study in Finnish adolescents by Felson and colleagues, where just over half of the association was found to be spurious.” But half of the association was “causal.” The results aren’t that inconsistent. And I think they included people who never drink. Also, the precision of the percentages indicated causality is unknown.
Clarify this statement: “…any prevention strategies for vandalism in young adults will need to consider the effects of intoxication and that these strategies may not be so useful if targeting people when they are sober.”
Reviewer 2 Report
This study examines whether alcohol has a causal effect on whether people engage in fighting, shoplifting, and vandalism. Using data from a longitudinal study of young adults in the UK, the authors estimate situational composition models to determine whether the difference between the total association between alcohol and offending is greater than the confounded association, indicating a potential causal effect. The findings suggested that alcohol had minimal causal effect on fighting and shoplifting, but a larger causal effect was observed for vandalism. The authors conclude crime prevention strategies for vandalism should not target people when they are sober.
The paper is, generally speaking, well written with just a few typos and grammatical errors. The logic of the paper is straightforward and fairly easy to follow. The study addresses the important topic of determining the causal effect of alcohol on offending behavior. They use a clever analytic technique to address causality with observational data. The authors are careful to point out, however, that inferring causation from observed correlations is difficult. I think this is critical because some readers will have serious concerns about claiming causation in this situation.
I have a couple concerns about the paper that I’ll detail now. First, I’m not sure what this study really adds to the literature on alcohol and crime, other than testing for the effects in a UK-based sample, whereas similar research has been conducted using Finnish and American samples. The authors should explain how their study differs from prior work, otherwise I’m not sure how this improves our understanding.
Second, the authors claim to use a “novel method for estimating causation with observational data” (line 479), but situational decomposition models are not new, as the authors clearly indicated earlier in the manuscript (e.g., line 115). A revised manuscript should not suggest that this method is new, unless it is somehow different from what Felson and colleagues have used in previous research. Perhaps using the multinomial logit model and comparing coefficients is the difference? A more detailed explanation is needed to differentiate the method from Felson et al. Also, I think the authors could explain the situational decomposition technique in a bit more detail. I had to go to the Felson et al. paper to really understand the logic of the approach. In particular, it just wasn’t explained in lines 100-103 why the association between alcohol and sober crime quantifies the effect of time stable confounders. A bit more explanation here would help the reader understand the approach so that they do not need to go back to Felson et al. to fully grasp the logic.
As a more minor point, some readers may wish to see a statistical test for the difference between the coefficients. I know the authors presented confidence intervals, but a formal test would solidify the difference.
Finally, I would say that the policy implications discussed at the very end of the paper are a bit weak. Are the authors suggesting in lines 476-477 that programs should target vandalism when people are not sober? I think the implication is that programs should address drinking problems, if being intoxicated is causing people to vandalize property. In any case, I think this argument could be improved a bit.
Overall, I like the paper and think the results should be published to contribute to this body of research.
